# Mumefural Improves Recognition Memory and Alters ERK-CREB-BDNF Signaling in a Mouse Model of Chronic Cerebral Hypoperfusion

**DOI:** 10.3390/nu15143271

**Published:** 2023-07-24

**Authors:** Min-Soo Kim, Bu-Yeo Kim, Jung Im Kim, Joungbok Lee, Won Kyung Jeon

**Affiliations:** 1KM Convergence Research Division, Korea Institute of Oriental Medicine, Daejeon 34054, Republic of Korea; kms3167@kiom.re.kr (M.-S.K.); buykim@kiom.re.kr (B.-Y.K.); imlee1001@kiom.re.kr (J.I.K.); 2Department of Biohealth Regulatory Science, Sungkyunkwan University, Suwon 16419, Republic of Korea; 3SONIMEDI Co., Incheon 21642, Republic of Korea; james531@sonimedi.com

**Keywords:** chronic cerebral hypoperfusion, cognitive impairment, mumefural, vascular dementia

## Abstract

Cognitive impairment resulting from chronic cerebral hypoperfusion (CCH) is known as vascular dementia (VaD) and is associated with cerebral atrophy and cholinergic deficiencies. Mumefural (MF), a bioactive compound found in a heated fruit of *Prunus mume* Sieb. et Zucc, was recently found to improve cognitive impairment in a rat CCH model. However, additional evidence is necessary to validate the efficacy of MF administration for treating VaD. Therefore, we evaluated MF effects in a mouse CCH model using unilateral common carotid artery occlusion (UCCAO). Mice were subjected to UCCAO or sham surgery and orally treated with MF daily for 8 weeks. Behavioral tests were used to investigate cognitive function and locomotor activity. Changes in body and brain weights were measured, and levels of hippocampal proteins (brain-derived neurotrophic factor (BDNF), extracellular signal-regulated kinase (ERK), cyclic AMP-response element-binding protein (CREB), and acetylcholinesterase (AChE)) were assessed. Additionally, proteomic analysis was conducted to examine the alterations in protein profiles induced by MF treatment. Our study showed that MF administration significantly improved cognitive deficits. Brain atrophy was attenuated and MF treatment reversed the increase in AChE levels. Furthermore, MF significantly upregulated p-ERK/ERK, p-CREB/CREB, and BDNF levels after UCCAO. Thus, MF treatment ameliorates CCH-induced cognitive impairment by regulating ERK/CREB/BDNF signaling, suggesting that MF is a therapeutic candidate for treating CCH.

## 1. Introduction

Vascular dementia (VaD) manifests as a progressive decline in cognitive function due to reduced blood supply to the brain [1]. The proportion of patients with dementia and VaD is estimated to increase by 40% over the next 30 years [2]. As VaD correlates with age, the need to develop cures in aging societies is growing.

Although the underlying etiology of VaD is poorly understood, common pathological characteristics include cerebral atrophy, neurodegeneration, and cholinergic deficiency [3,4]. Several researchers are attempting to find effective drugs for VaD; however, currently, no FDA-approved treatments for VaD exist. To date, some cholinesterase inhibitors used for Alzheimer’s disease (AD) and herbal preparations, including the Ginkgo biloba extract, have been used to treat VaD [5,6]; however, further integrated and effective anti-VaD drugs are required.

Herbal medicines have long been used throughout Asia, and >80% of people worldwide rely on herbal medicines for healthy living [7]. Recent advances in science and technology have facilitated the bioactive compound isolation from plants. Mumefural (MF) is one of the bioactive compounds found in the heated fruit of *Prunus mume* Sieb. et Zucc [8]. According to previous studies, *P. mume* extract and MF have beneficial effects on blood fluidity, especially by inhibiting platelet aggregation [9,10]. Thus, we have noted their therapeutic potential in treating VaD. The extracts of *Fructus mume*, a processed fruit of *P. mume*, help slow the progression of cognitive impairment induced by chronic cerebral hypoperfusion (CCH), which is a major cause of VaD [11,12]. Additionally, our recent study found that MF treatment prevented spatial memory deficits and neurological dysfunction and inhibited neuroinflammation in a rat CCH model [13]. These results imply that developing MF as a potential treatment for VaD is valuable; however, further investigation using various animal models is still needed to confirm whether MF protects the VaD brain. Here, we assessed the reproducibility of the beneficial effects of MF in an animal model of VaD mice subjected to unilateral common carotid artery occlusion (UCCAO). Furthermore, we conducted various behavioral tasks to measure different memory types (e.g., working memory and object recognition memory) and their underlying molecular mechanisms. Finally, we investigated whether MF administration affects brain atrophy induced by CCH and performed proteomic analysis to unravel the potential MF molecular mechanism.

## 2. Materials and Methods

### 2.1. Subjects

Thirty C57BL/6 male mice were purchased from OrientBio Co., (Seongnam, South Korea). The mice were 8 weeks old at the time of arrival and were supplied with food and water ad libitum for two weeks before initiating the experiments. They were housed in groups with three to four animals per cage and controlled with a 12:12 h light–dark cycle, humidity (50% ± 10%), and temperature (22 ± 1 °C). All procedures were conducted according to protocols approved by the Institutional Animal Care and Use Committee of the Korea Institute of Science and Technology (approval number: 2021-07-085).

### 2.2. Experimental Design

Mice were randomly assigned into three groups after acclimatization (*n* = 10 per group): (1) sham-operated + vehicle (SHAM + VEH), (2) UCCAO + vehicle (UCCAO + VEH), and (3) UCCAO + MF. MF was obtained from U CHEM (Anyang, South Korea). MF was dissolved in saline, which was used as the vehicle. Two weeks after the surgery, the vehicle or MF (40 mg/kg) was administered daily via oral gavage for two months. The doses selected were based on those used in our previous study [13]. Nine weeks after surgery, all the mice performed behavioral tasks. The mice were sacrificed 10 weeks after surgery (refer to Figure 1 for the schedule of this study).

### 2.3. Surgery

UCCAO was performed as previously described [14]. Briefly, after a 2 cm midline neck incision under anesthesia with 2% isoflurane, the right common carotid artery was carefully isolated and double-ligated using 6–0 silk sutures. The same procedure was performed in the sham mice, except that the arteries were not occluded. The incision was closed using layer-by-layer sutures and disinfected with povidone-iodine. After 30 min of monitoring, the mice were returned to their original cages.

### 2.4. Behavioral Tasks

All behavioral tasks were conducted 9–10 weeks after surgery in the following order: Y-maze, open field (OF), and novel object/location recognition tests (NOR/NLR). The mice were allowed to adapt to the hands of the experimenter for 3 min/d over 5 consecutive days before the tasks.

#### 2.4.1. Y-Maze Test

The Y-maze test was performed to evaluate short-term working memory [14]. Each mouse was placed at the junction of three identical arms (40 × 10 × 15 cm; length–width–height) and allowed to freely explore for 8 min. All the trials were recorded and analyzed with the number of arm entries and alternations. Spontaneous alternation (%) was defined according to the following equation:$$Spontaneous\;alternation\left(\%\right)=\frac{Actual\;number\;of\;alternations}{Total\;number\;of\;arm\;entries-2}\times 100$$

#### 2.4.2. Open Field Test

The open field test was used to assess general locomotor activity [15]. Mice were individually placed in the central area of the open field box (40 × 40 × 40 cm; length–width–height) and acclimatized for 10 min. The movements of the mice were recorded by the camera. The total distance moved (cm) and the time spent in the center 24 × 24 cm squared zone were measured using a behavioral analysis software (HVS image, Hampton, UK, https://hvsimage.com/, accessed on 2 July 2023).

#### 2.4.3. NOR/NLR Test

A NOR/NLR test was performed as previously described to evaluate episodic memory [16]. The protocol consisted of four phases: (1) habituation, (2) sampling, (3) novel object, and (4) novel object location. First, each mouse was individually allowed to explore for 10 min in the open field box without objects. The next day, in the sample phase, two identical objects (A and A’) were presented, and each animal was allowed to explore the objects for 10 min. In the NOR phase, the mice were allowed to explore one identical sample object (A or A’) and a novel object (B) for 5 min, and the time spent exploring each object was scored for 3 min after 24 h of the sample phase. In the NLR phase, one of the objects previously presented was moved to a new location in the arena, and the mice were allowed to freely explore for 5 min. The exploration time for each object was measured through exploratory behavior, such as licking, touching, or sniffing the object, excluding sitting on the object. Data were excluded from the analysis if the mice explored the objects for <5 s. The recognition index was calculated according to the following equation:$$Recognition\;index=\frac{Time\;spent\;exploring\;new\;object\;(or\;new\;location)}{Total\;time\;spent\;exploring\;both\;obejcts}$$

### 2.5. Brain Sample Preparation

All mice were euthanized by decapitation 10 weeks after surgery. The brain tissues of decapitated mice were rapidly isolated and weighed. Each brain tissue was cut into two hemispheres (ipsilateral and contralateral) and weighed individually. The ipsilateral part of the brain was dissected to collect the hippocampus and cerebellum and kept at −80 °C for further experiments.

### 2.6. Western Blotting

Western blotting was performed according to previous research [17]. Each hippocampal tissue sample was homogenized in radio-immunoprecipitation assay buffer with protease inhibitors (GenDEPOT, Baker, TX, USA) and centrifuged at 14,000× *g* for 1 h. Protein concentrations were quantified using a bicinchoninic acid assay kit (Thermo Fisher Scientific, Waltham, MA, USA), and 20 µg of total protein was separated using 10% sodium dodecyl sulfate polyacrylamide gel electrophoresis. The resolved proteins were transferred to polyvinylidene fluoride membranes, which were blocked with fat-free milk in Tris-buffered saline containing 0.1% (*v*/*v*) Tween-20 (TBST). Subsequently, the membranes were incubated overnight at 4 °C with the primary antibodies. After extensive washing with TBST, the membranes were incubated with horseradish-peroxidase-conjugated anti-secondary antibodies (Cell Signaling Technology, Danvers, MA, USA) for 1 h at room temperature (23 ± 2 °C). Immunosignals were visualized using a chemiluminescence system and quantified using Image Gauge (Fujifilm, Minato-ku, Tokyo, Japan). The band intensity of the sham-operated + vehicle group was assigned a value of 100, and the relative levels of each sample were presented. Detailed information on the primary antibodies is presented in Appendix A.

### 2.7. Protein Digestion and Liquid Chromatography–Mass Spectrometry (LC-MS)/MS Analysis

Proteomic analyses were performed as described previously [18]. Briefly, hippocampal tissues were sonicated and digested with trypsin at 37 °C, followed by inactivation with formic acid (Honeywell, Dallas, TX, USA). C18 spin columns (Harvard Apparatus, Holliston, MA, USA) were used to desalt the peptides. Samples were analyzed by LC–MS/MS. Quantification of protein was performed and data were analyzed using Mass Profiler Professional (Agilent Technologies, Santa Clara, CA, USA). The detailed methodology is presented in the Appendix A.

### 2.8. Isolating Differentially Expressed Proteins

Differentially expressed proteins whose expression differs for each experimental group compared to the control group were statistically selected using a protein interaction network. BioGRID was used as the protein interaction database [19]. Specifically, for each sample, adjacent proteins that directly interact with the protein of interest were selected as modules, and then the average and standard deviation of these modules were measured. To determine the statistical significance of these values, they were compared with values obtained from 1000 randomly selected modules of the same size. For comparison between samples, the Pearson correlation coefficient between modules was additionally measured, and statistical significance was measured in the same way using random permutation.

### 2.9. Functional Network of Gene Ontology (GO) Terms

The ClueGO and DAVID programs (https://david.ncifcrf.gov, accessed on 6 March 2023) were used to extract GO terms associated with differentially expressed proteins and to measure the network connectivity structure of these GO terms [20,21].

### 2.10. Measurement of Pathway Activity

As we previously reported, pathway activity levels were measured from the expression levels of individual proteins constituting the pathway [22]. In detail, first, information on individual pathways and their constituent proteins was obtained from the KEGG database. Then, the total sum of log expression levels compared to the control group of proteins included in each pathway was measured and used as the activity value. In this process, the weight of −1 was multiplied by the expression level for the proteins that suppressed the signal transduction of the pathway and then added together to reflect the effect of suppressing the signal transduction of these repressor proteins. To obtain the statistical significance of the measured pathway activity values, proteins of the same size for each pathway were randomly extracted 1000 times, the above process was repeated, and the measured values and pathway activity values were compared.

### 2.11. Statistical Analysis

We presented all data as mean ± standard error of the mean (SEM). The homogeneity of variances was evaluated using Levene’s test. Pearson’s correlation analysis was applied to investigate correlations between protein expressions and the recognition index, and other between-group comparisons were analyzed using one-way analysis of variance (ANOVA) followed by the least significant difference test. *p*-value < 0.05 was considered statistically significant.

## 3. Results

### 3.1. Effects of MF on the Body Weight

Mice were subjected to sham or UCCAO surgery for induction of vascular dementia (VaD) after a 2-week adaptation period (Figure 1). We measured the body weights of the mice weekly from before to 10 weeks after surgery. UCCAO mice showed significantly increased body weight compared to sham mice at 10 weeks after surgery (5.10 ± 0.31 g in SHAM + VEH; 6.60 ± 0.37 g in UCCAO + VEH), consistent with previous reports [15,23]. However, MF administration successfully reduced the UCCAO-induced body weight gain (4.70 ± 0.47 g in UCCAO + MF, *p* < 0.01; Figure 2).

### 3.2. Effects of MF on the Cognitive Impairment Induced by CCH

We performed an open field test to assess locomotor activity and anxiety levels in mice. The differences in the total distance moved and center time during a 15 min period in the open field apparatus were insignificant (Figure 3A–C), indicating no obvious defects in locomotion or anxiety in any of the mice. The Y-maze test was employed to study MF effects on spontaneous working memory. Total arm entries did not vary among the experimental groups (*p* > 0.05; Figure 3D), indicating that UCCAO surgery and MF administration did not affect general exploratory behavior. The UCCAO group exhibited a notable reduction in the spontaneous alternation rate compared with the sham group (*p* < 0.05), suggesting impaired working memory in the UCCAO mice. However, the differences between the UCCAO + VEH and UCCAO + MF mice were insignificant (*p* > 0.05; Figure 3E). The NOR/NLR test was employed to evaluate MF effects on episodic memory. All groups spent similar amounts of time exploring the same objects during the sampling phase (Figure 3F–G), indicating that UCCAO and MF did not affect motivation or innate exploratory behavior. In the NOR phase, the UCCAO + VEH group spent less time exploring new objects than the SHAM + VEH group (*p* < 0.001); however, this discrimination ratio was recovered in the UCCAO + MF mice relative to the UCCAO + VEH mice (*p* < 0.05; Figure 3H). Additionally, UCCAO mice showed impaired recognition memory in the NLR phase (*p* < 0.05), and an increasing trend in the recognition index was observed in the UCCAO + MF mice, although statistically insignificant (*p* = 0.120; Figure 3H). Thus, these results suggest that MF improves UCCAO-induced recognition memory loss.

### 3.3. Effects of MF on the Brain Weight

Because brain atrophy is clinically known to be related to chronic cerebral hypoperfusion (CCH) [24], we measured the brain weight of each group. UCCAO significantly decreased the weight of the whole brain, ipsilateral ischemic hemisphere (*p* < 0.001), and ipsilateral hippocampal tissue (*p* < 0.05) in the other groups compared with that in the sham group (Figure 4). No detectable changes were observed in the weights of the cerebellum, contralateral hemisphere, or hippocampus in the UCCAO group. MF administration tended to suppress the decrease in the weights of the brains and ipsilateral hippocampus in mice subjected to UCCAO, although the difference was insignificant. These results suggested that MF has an impact on increasing brain weight, especially in the hippocampus, in the UCCAO model.

### 3.4. Effects of MF on Glucose Transporter (GLUT) Expression Induced by CCH

Given that GLUT expression is elevated in response to CCH [23], Western blot analysis was performed to investigate MF effects on GLUT expression in the ipsilateral hippocampus. A remarkable increase was found in the expression of brain-type GLUTs (GLUT1 and GLUT3) in the UCCAO mice compared with that in the sham mice (GLUT1: *p* < 0.01; GLUT3: *p* < 0.05). Interestingly, MF treatment notably down-regulated the levels of these proteins (GLUT1: *p* < 0.05; GLUT3: *p* < 0.05). The protein levels of liver-type GLUT2, which was used as a negative control, did not differ among the groups (*p* > 0.05; Figure 5A). These results suggested that MF modulates CCH-induced GLUT expression.

### 3.5. Effects of MF on Acetylcholinesterase (AChE) Levels Induced by CCH

Next, we measured the levels of AChE in each group, because cholinergic dysfunction is known to be involved in cognitive impairment caused by CCH [25]. The level of AChE was markedly increased in the UCCAO group compared to the sham group (*p* < 0.01; Figure 5B), and MF administration down-regulated the levels of AChE (*p* < 0.05). These findings imply that MF regulates CCH-induced cholinergic alterations.

### 3.6. Effects of MF on Protein Expression Involved in Learning and Memory Pathways

Brain-derived neurotrophic factor (BDNF) is a key regulator for maintaining neural functions, including memory [26]; thus, we investigated the levels of BDNF and its receptor, tropomyosin receptor kinase B (TrkB), in the hippocampus using Western blotting. BDNF level was notably reduced in the UCCAO + VEH group compared to the SHAM + VEH group (*p* < 0.05; Figure 6A); however, MF administration considerably rescued the BDNF downregulation induced by UCCAO (*p* < 0.05). The differences in phosphorylation levels of TrkB, normalized to total TrkB levels, among the groups were insignificant (*p* > 0.05; Figure 6A).

BDNF triggers the activation of extracellular signal-regulated protein kinase (ERK) and its downstream target, cAMP-response element binding protein (CREB), in the hippocampus [27]. To further examine the underlying mechanisms, the levels of ERK and CREB as well as their phosphorylation in the hippocampus were investigated. UCCAO reduced the expression levels of p-CREB and p-ERK compared to that in the sham group (Figure 6B; p-CREB/CREB: *p* < 0.05; p-ERK/ERK: *p* < 0.01); however, MF significantly restored the reduction in ERK and CREB phosphorylation induced by UCCAO (p-CREB/CREB: *p* < 0.01; p-ERK/ERK: *p* < 0.05).

Subsequently, a Pearson’s correlation analysis was used to investigate whether the alteration in ERK-CREB-BDNF signaling correlated with cognitive performance on the recognition index in the NOR test. The recognition index was significantly and positively correlated with BDNF (R^2^ = 0.3402; *p* < 0.01), p-ERK (R^2^ = 0.4029; *p* < 0.01), and p-CREB levels (R^2^ = 0.1645; *p* < 0.05; Appendix A). Overall, these results suggest that MF administration is capable of preventing the attenuation of ERK-CREB-BDNF signaling found in the hippocampus of UCCAO mice, and MF activating this signaling is partly associated with the cognitive enhancing effect.

### 3.7. Effects of MF on Proteomic Profiling

We measured the effect of MF on protein expression in UCCAO mice. The levels of several proteins were altered in the UCCAO mice, whereas the levels of other proteins were altered by MF administration (Figure 7A). Additionally, these MF effects were observed in a pathway activity analysis, in which UCCAO and MF administration altered several pathways with various functions (Appendix A). Subsequently, we isolated the proteins that were differentially expressed in each sample or between UCCAO + VEH- and UCCAO + MF-treated samples. We used a protein interaction network rather than a simple threshold to reduce the errors in the selected proteins in this process. The distribution of the differentially expressed proteins indicated that the target proteins of MF differed from those of UCCAO (Figure 7B). Additionally, the biological function association analysis of these proteins supported the functional differences induced by UCCAO and MF treatment (Figure 7C and Appendix A). For example, pathways associated with neurodegeneration, such as AD, Huntington’s disease (HD), and Parkinson’s disease (PD), were enriched in UCCAO. Additionally, oxidative phosphorylation during aerobic respiration is involved in UCCAO. However, the target functions induced by MF mainly consist of metabolic functions, such as amino acid biosynthesis, carbon metabolism, nucleoside metabolism, and heme (or porphyrin) metabolism.

## 4. Discussion

Here, we investigated the effects of MF on CCH-induced cognitive impairment and its underlying mechanisms using a UCCAO mouse model. MF resolved CCH-associated pathological changes, including recognition memory impairment and brain atrophy. In our previous study, orally administering up to 5000 mg/kg/d of MF showed no significant acute or subacute toxicity [28]. Considering these findings, MF may be a favorable and safe therapeutic candidate for treating VaD.

Previously, we demonstrated that MF effectively recovered cognitive impairment and neuroinflammation induced by bilateral common carotid artery occlusion (BCCAO), a widely accepted VaD rat model [13]. However, unlike rats, which have a fully developed circle of Willis, mice have relatively less-developed posterior communicating arteries; thus, permanent BCCAO can cause death in them [29]. Therefore, several researchers have used the UCCAO mice model, which mimics the CCH condition of VaD. Among the pathological characteristics of UCCAO, increased body weight resulting from the dysregulation of metabolism has been reported [23]. Similar to the results of previous reports, our results showed that the UCCAO group gained more weight than the sham group and that MF treatment alleviated body weight gain in the UCCAO group. Additionally, we observed UCCAO-induced brain atrophy, particularly in the ipsilateral hippocampus, which was partially ameliorated by MF administration. Previous reports have suggested that reduced cerebral blood flow is associated with brain atrophy and that the hippocampus is particularly vulnerable to CCH in several brain regions [30,31]. Our results support previous findings that the UCCAO model of CCH causes hippocampal atrophy [24]. Furthermore, similar to previous observations, the expression of GLUT1 and GLUT3, which are involved in brain glucose metabolism, was upregulated after UCCAO to compensate for decreased glucose utilization following CCH; however, liver-type GLUT2 was not altered in the present study [15]. Interestingly, MF treatment markedly decreased the UCCAO-induced GLUT upregulation. Thus, we postulated that the beneficial effect of MF on UCCAO is related to glucose metabolism regulation; however, further studies are required to examine whether this effect is direct or indirect.

VaD-related cognitive impairment is commonly observed in animal models and humans. Among the several memory types, patients with VaD have episodic memory deficits, which involve conscious recollection of individual events [32,33]. UCCAO mice have previously shown poor performance in the NOR test, which is a valuable tool for assessing episodic-like memory in rodents [23,24]. In the present study, UCCAO mice exhibited deficits in the NOR test, together with the Y-maze and NLR tests, and MF administration rescued the deficit in the NOR test. To the best of our knowledge, this is the first study to investigate the effect of MF on recognition memory in a VaD rodent model.

The cholinergic system has been reported to be vulnerable to vascular injury, and this cholinergic deficiency is noticeable in patients with VaD [34]. Because a lack of cholinergic neurotransmitters can lead to cognitive impairment, the inhibition of excessive acetylcholine hydrolysis can help improve cognition [35]. Therefore, cholinesterase inhibitors have been used to slow disease progression. In the present study, AChE levels in the hippocampus were significantly lower in the MF-treated mice than that in the UCCAO mice, indicating that MF modulates cholinergic mechanisms. However, further experiments are required to determine whether MF affects cholinergic transmission via its transporters.

ERK is involved in cell growth and signal transmission and is thought to play an important role in cognition [36]. Activated ERK induced by its phosphorylation is linked to CREB phosphorylation and subsequently enhances its transcriptional target, the BDNF gene, which is required for maintaining hippocampus-dependent long-term memory [37,38]. UCCAO decreased ERK and CREB phosphorylation and BDNF levels in the hippocampus, whereas chronic MF treatment ameliorated these alterations. Alterations in BDNF levels were not accompanied by significant changes in its receptor, tyrosine kinase B (TrkB). A previous study showed that hippocampal atrophy is accompanied by decreased BDNF expression in patients with depression [39]. Similarly, we observed brain atrophy and a reduction in BDNF levels in UCCAO mice, suggesting that BDNF may be closely linked to brain atrophy. BDNF is mainly involved in neuronal development, synaptic plasticity, and cognitive function. When BDNF binds its receptor TrkB, several molecular signaling pathways, including phosphoinositide 3-kinase, and phospholipase Cg are initiated [40]. A recent study has shown that reduced serum BDNF is associated with Alzheimer’s disease (AD) pathology and can be used as a biomarker for the detection of AD [41]. Although the role of BDNF in VaD is not as well researched as in AD, several studies using VaD models have reported BDNF-related changes. Liu et al. revealed that decreased levels of BDNF and neuronal cell death were shown in UCCAO mice and were reversed by probiotics [42]. Taken together, these studies and our results suggest that BDNF may be a promising target for the treatment of VaD.

Additionally, we analyzed the correlation between ERK/CREB/BDNF signaling and recognition memory. Positive correlations were observed in the hippocampus, with a higher recognition index associated with higher p-ERK, p-CREB, and BDNF levels. These results imply that the memory-enhancing effect of MF may be modulated by the activation of ERK/CREB/BDNF signaling in the hippocampus; however, future studies are required to decipher the causal relationship.

Finally, we investigated the changes in the hippocampal proteomes of the experimental groups. Several pathways, including those of individual proteins, were notably different between the UCCAO + VEH and UCCAO + MF groups. Aerobic respiration was detected in the UCCAO group in the GO term enrichment analysis. Prior studies have reported that aerobic respiration can be regulated by glucose metabolism and that its critical factor contains GLUTs [43,44]. Similarly, our results showed that GLUT expression was significantly upregulated under UCCAO conditions, and that MF treatment ameliorated GLUT expression. Interestingly, the pathways identified in UCCAO are mainly involved in neurodegenerative diseases, such as AD, HD, and PD. Targeting ERK/CREB/BDNF signaling has been suggested as a therapeutic strategy for these diseases [45,46]. Considering our findings, future studies on the treatment effects of MF on other neurodegenerative diseases are warranted.

There are several limitations to our study. First, we used only male mice for this study, in line with previous research that shows that VaD occurs more often in men than in women [47]. However, many women are also suffering from VaD in aging societies. Therefore, there is a need for future research to substantiate the efficacy of MF using female animals. Second, we did not include an MF-treated sham group in the present study. To confirm whether MF has memory-enhancing effects in the naïve animal, further studies including a sham + MF group would provide a better understanding of the effects of MF. Third, we have studied the effect of MF at one dose (40 mg/kg) in the present experiments, so increasing the range of doses of MF could achieve more improvements against CCH, which should be investigated. Lastly, we focused on changes in proteins in this study; changes in the transcriptome using RNA-seq caused by MF treatment need to be investigated.

## 5. Conclusions

In conclusion, CCH stimulates a series of VaD symptoms, including cognitive impairment, brain atrophy, and biochemical alterations in a UCCAO mouse model. MF administration conferred beneficial effects in UCCAO mice, including the reversal of episodic memory impairment. Thus, the underlying mechanisms may involve the ERK/CREB/BDNF signaling pathway. These findings underline the promising use of MF as a treatment option for VaD.

## Figures and Tables

**Figure 1 nutrients-15-03271-f001:**
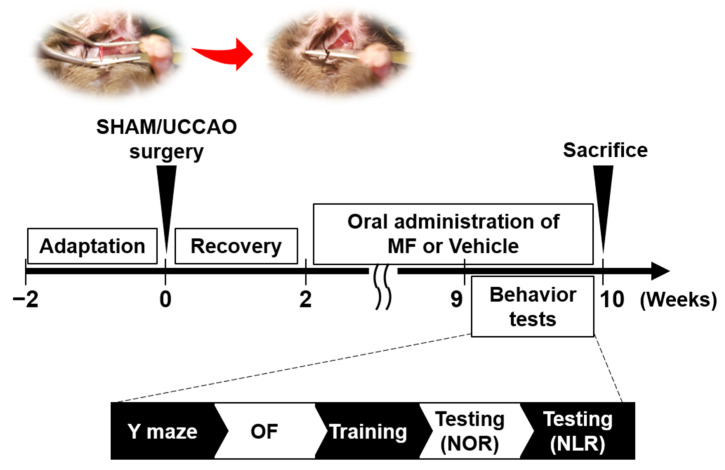
Study design. Mice were subjected to sham or UCCAO surgery after 2 weeks of acclimatization. They were orally treated with MF or vehicle (saline) for two months following the recovery period. Behavioral tests were conducted between 9 and 10 weeks after surgery, and all mice were sacrificed 10 weeks after surgery. Brain samples were collected for further analyses. VEH, vehicle; MF, mumefural; NOR, novel object recognition test; NLR, novel location recognition test; OF, open field test.

**Figure 2 nutrients-15-03271-f002:**
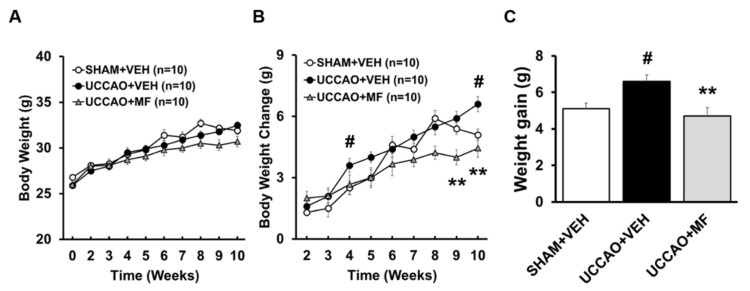
Effects of MF on body weight. All mice were weighed every week. The body weight (**A**), changes in body weight (**B**), and body weight gain (**C**) are shown. The number in parentheses denotes the number of samples. Each point represents mean ± SEM. # *p* < 0.05 compared with SHAM + VEH; ** *p* < 0.01 compared with UCCAO + VEH. VEH, vehicle; MF, mumefural.

**Figure 3 nutrients-15-03271-f003:**
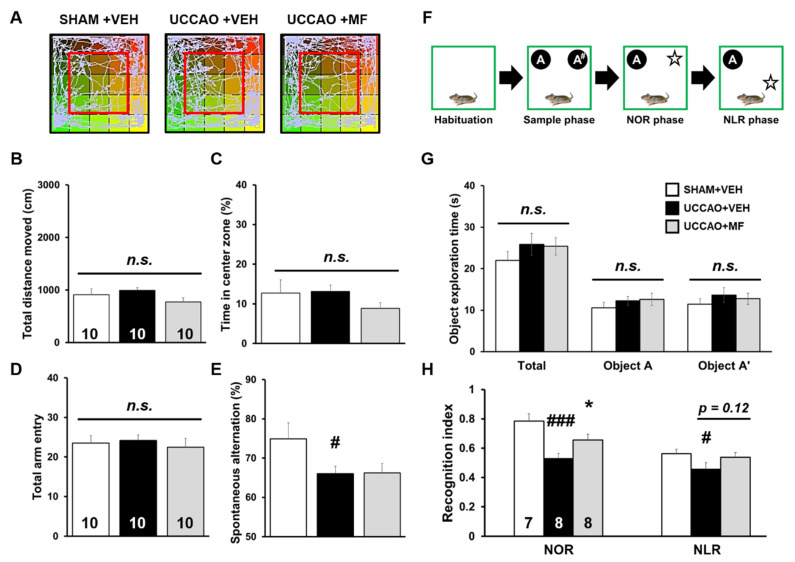
Effects of MF on general behavior and cognition. (**A**–**C**) Locomotor behavior was assessed using an open field test. (**A**) Representative movement tracking of each group in the open field. The inner bold box indicates the center zone of the open field. Total distance that the mice moved (**B**) and the time spent in the center (**C**) were examined in the open field. (**D**,**E**) Short-term working memory was assessed in a Y-maze test. Total number of arm entries (**D**) and the calculated rate of spontaneous alternation (**E**) for 8 min in the Y-maze test. (**F**–**H**) Episodic memory was assessed in a novel object/location recognition (NOR/NLR) test. (**F**) Behavioral procedure for the (NOR/NLR) test. Time spent exploring during the sample phase (**G**) and recognition index during two test phases of the test (NOR and NLR phases) (**H**). The numbers shown in the bar group correspond to the number of samples included in the analysis. Data are presented as mean ± SEM. Statistical significance was indicated as follows: # *p* < 0.05, and ### *p* < 0.001 when compared with the SHAM + VEH group. * *p* < 0.05 compared with the UCCAO + VEH group. VEH, vehicle; MF, mumefural; n.s., not significant.

**Figure 4 nutrients-15-03271-f004:**
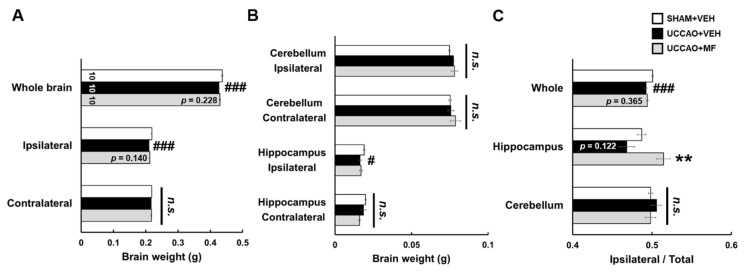
Effects of MF on brain weight. (**A**) Weight of the whole brain and ipsilateral/contralateral ischemic hemisphere in each group. (**B**) Weight of the cerebellum and hippocampal tissue in the ipsilateral/contralateral sides. (**C**) Weight ratio of the ipsilateral side to the total in the whole, hippocampus, and cerebellum tissue. The numbers shown in the bar group correspond to the number of samples included in the analysis. Data are presented as mean ± SEM. Statistical significance was indicated as follows: # *p* < 0.05 and ### *p* < 0.001 when compared with the SHAM + VEH group; ** *p* < 0.01 compared with UCCAO + VEH group. VEH, vehicle; MF, mumefural; n.s., not significant.

**Figure 5 nutrients-15-03271-f005:**
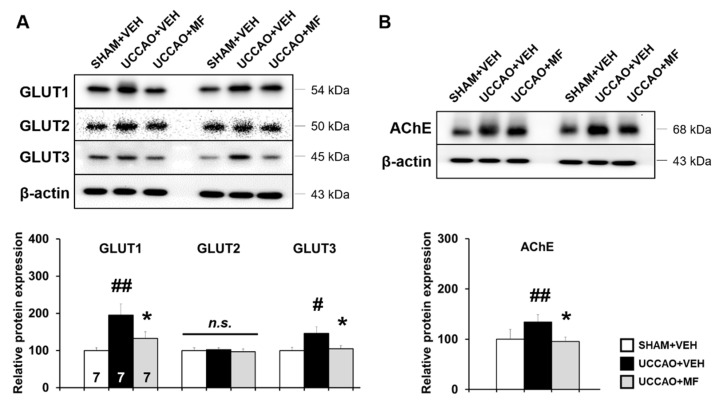
Effects of MF on the glucose transporters (GLUT) and acetylcholinesterase (AChE) levels. (**A**) Upper: the GLUT1, GLUT2, and GLUT3 levels in the hippocampus. Lower: GLUT quantification. (**B**) Upper: AChE levels in the hippocampus. Lower: AChE quantification. Bands were normalized to β-actin loading control with the SHAM + VEH group value set as 100. The resulting bar graph is presented as the percentage relative to the SHAM + VEH group. The numbers shown in the bar group correspond to the number of samples included in the analysis. Data are presented as mean ± SEM. Statistical significance was indicated as follows: # *p* < 0.05 and ## *p* < 0.01 when compared with the SHAM + VEH group. * *p* < 0.05 compared with the UCCAO + VEH group. VEH, vehicle; MF, mumefural; n.s., not significant.

**Figure 6 nutrients-15-03271-f006:**
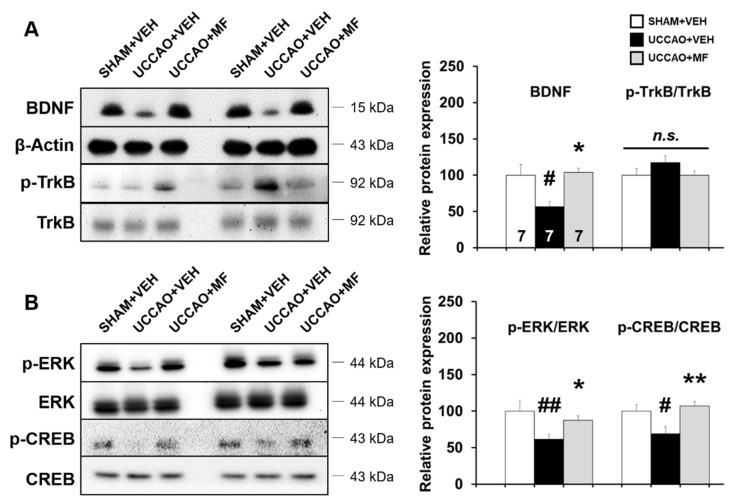
Effects of MF on the levels of ERK/CREB/BDNF signaling. (**A**) Left: the BDNF and TrkB levels in the hippocampus. Right: BDNF and p-TrkB/TrkB quantification. (**B**) Left: ERK and CREB levels in the hippocampus. Right: p-ERK/ERK and p-CREB/CREB quantification. Bands were normalized to β-actin loading control with the SHAM + VEH group value set as 100. The resulting bar graph was presented as the percentage relative to the SHAM + VEH group. The numbers shown in the bar group correspond to the number of samples included in the analysis. Data are presented as mean ± SEM. Statistical significance was indicated as follows: # *p* < 0.05 and ## *p* < 0.01 when compared with the SHAM + VEH group. * *p* < 0.05 and ** *p* < 0.01 compared with the UCCAO + VEH group. VEH, vehicle; MF, mumefural; n.s., not significant.

**Figure 7 nutrients-15-03271-f007:**
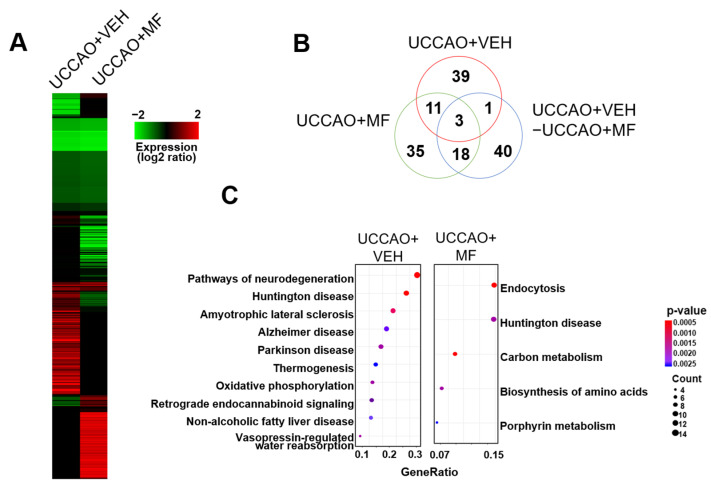
Effects of MF on protein expression in UCCAO mice. (**A**) In the experimental group, 1188 proteins showing a difference in the expression level of more than 1.5 times compared to the control group were selected, and clustering analysis was performed according to their expression level pattern. As shown in the scale bar at the bottom, proteins with increased or decreased expression are indicated in red or green, respectively. (**B**) After isolating the differentially expressed proteins in each sample or between samples (UCCAO-MF), their distributions were compared. (**C**) Functions of the differentially expressed proteins in the samples were measured in the pathway.

## Data Availability

The data presented in this study are available in this article. Further inquiries can be directed to the corresponding author.

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
