# Peer review of "Mumefural Improves Recognition Memory and Alters ERK-CREB-BDNF Signaling in a Mouse Model of Chronic Cerebral Hypoperfusion"

_nutrients, 2023, doi:10.3390/nu15143271_

Round 1
Reviewer 1 Report
This manuscript investigated the protective effect of mumefural in a mouse model of chronic cerebral hypoperfusion. They demonstrated that MF administration significantly improved cognitive deficits and the protective mechanism is associated with ERK/CREB/BDNF signaling. The authors also used proteomic analysis to examine the changes in protein expression profiles induced by MF treatment. Overall the manuscript is interesting and the results suggested that MF is a therapeutic candidate for treating CCH. Some minor concerns:
1. It seems that the protective effect of MF is moderate. Is it possible to increase the dose to get more improvement in cognitive deficits?
2. It seems that MF had some effects on the brain weight. Is it possible to provide the H&E or other kinds of staining of brain sections to visualize the changes?
3. Did the changes in ERK/CREB/BDNF signaling also be observed in the proteomic analysis results. It is better to validate some critical changes in protein expression via Western blot.
Author Response
Dear editor and reviewers,
We are greatly thankful for your kind comments and suggestions on our manuscript (Manuscript ID: nutrients-2512814). There is no doubt that these comments and suggestions are valuable and very helpful for revising and improving our manuscript. We have studied the comments carefully and have made corrections which we hope are met with approval. Changes to our manuscript are in red font or highlighted in yellow in the revised manuscript. Herein, we would like to respond to your comments and give a detailed account of the changes made to the original manuscript.
We are very grateful for your hard and warm work in reviewing our manuscript. Thank you again for your positive and constructive comments and suggestions. We earnestly hope you will find our revised manuscript acceptable for publication.
Yours sincerely,
Reviewer 1
This manuscript investigated the protective effect of mumefural in a mouse model of chronic cerebral hypoperfusion. They demonstrated that MF administration significantly improved cognitive deficits and the protective mechanism is associated with ERK/CREB/BDNF signaling. The authors also used proteomic analysis to examine the changes in protein expression profiles induced by MF treatment. Overall the manuscript is interesting and the results suggested that MF is a therapeutic candidate for treating CCH. Some minor concerns:
Response: Thank you for the detailed review of our manuscript. Our responses are detailed below.
- It seems that the protective effect of MF is moderate. Is it possible to increase the dose to get more improvement in cognitive deficits?
Response: We agree perfectly with this remark. We previously investigated the effects of MF in the rat CCH model, established by bilateral common carotid arterial occlusion (BCCAO) (Bang, J., Kim, M. S., & Jeon, W. K. (2019). Mumefural ameliorates cognitive impairment in chronic cerebral hypoperfusion via regulating the septohippocampal cholinergic system and neuroinflammation. Nutrients, 11(11), 2755.). At that time, rats were orally administered saline or MF at 20, 40, or 80 mg/kg body weight, and MF treatment attenuated BCCAO-induced pathological markers at doses of 40 and 80 mg/kg. Therefore, we chose the lower effective dose of 40 mg/kg according to a previous study. However metabolic processes between rats and mice may be different, so it will be valuable to examine the effects of MF at a higher dose. We have included this commentary in the discussion on page 13, lines 447-449.
Adopted from Nutrients 2019, 11(11), 2755
- It seems that MF had some effects on the brain weight. Is it possible to provide the H&E or other kinds of staining of brain sections to visualize the changes?
Response: We appreciate the reviewer’s suggestion. However, in the present study, all brain samples were dissected to collect hippocampal regions for western blotting and proteomics, so we do not have samples to experiment with staining. Because the brain weight changed, it is possible that morphological alterations occurred within brain subregions. Therefore, as suggested by the reviewer, we plan to perform histochemical analysis to study the association with changes in brain weight.
- Did the changes in ERK/CREB/BDNF signaling also be observed in the proteomic analysis results. It is better to validate some critical changes in protein expression via Western blot.
Response: We really appreciate the better suggestion of the reviewer. The proteome was analyzed in the present study, and about 2000 proteins were detected in the sample. Unfortunately, however, ERK, CREB, and BDNF proteins were not detected. Proteomics technologies are useful analysis systems to investigate biological function, but in the cases of the brain, neuroproteomics has probably not yet resulted in the detection of the majority of the important brain proteins due to low abundance and hydrophobic properties (Reference A). For that reason, it is sometimes difficult to see results from Western blots in proteomics. However, we first showed that there were functional differences (e.g. neurodegeneration, oxidative phosphorylation) induced by UCCAO and MF treatment in the proteomic analysis. Based on the present studies, we plan to perform other kinds of omics, such as RNA-seq, to detect a wider range of targets in the future. We have included this limitation in the discussion on page 13, lines 449-451.
#Reference A: Garbis, S.; Lubec, G.; Fountoulakis, M. Limitations of current proteomics technologies. J. Chromatogr. A, 2005, 1077, 1-18.

Reviewer 2 Report
Materials and Methods:
The biggest issue with the paper as it currently stands is the use of only males. In order to publish the current data the authors need to include stipulations with all of their conclusions regarding the translational capacity to move from the current results to the sphere of human health. This is especially egregious with regards to mention of particular human diseases, for example Alzheimer's, where the human presentation is not only sex-biased, but far more prevalent in females than males. Editing to make these limitations clear is needed. This applies to all sections, not just Materials and Methods.
Justification is needed for not including a sham+MF group. If anything is to be examined as a potential treatment, it is imperative to know how it behaves on its own.
Additionally, justification is needed for not counterbalancing the order of the behavioral tests.
Clarification is needed of if both the ipsilateral and contralateral hippocampi and cerebella were kept, or if only one side was kept.
Results:
All graphs and figures require editing - the current size of the text is extremely hard to read. Figure 7 requires even more editing, as even with increased text size, it is likely to contain too much information to be clearly visible. Should potentially be split between having the relevant information in the paper, and the rest in the supplemental data.
Regarding the Western Blots - why are there two sets of columns?
Line 18 - 'the' behavioral tests - 'the' is unnecessary.
Line 21 - 'were', not 'was'.
Reviewer 3 Report
The manuscript submitted by Kin-Soo Kim and colleagues entitled “Mumefural improves cognitive impairment in a mouse model of chronic cerebral hypoperfusion via regulation of ERK-CREB-3 BDNF signalling” reports the effects of a heated extract obtained from Prunus mume on the brain biochemistry after unilateral common carotid artery occlusion (UCCAO) in the mouse model.
Here are my comments:
Title:
I think the title is misleading as the only cognitive function improved by the extract was the recognition index and the rest of the data report enzyme expression and modification levels. Maybe a better title is “Mumefural improves curiosity in a mouse model of chronic cerebral hypoperfusion and impacts on ERK-CREB-3 BDNF signalling”. The data do not support a causative link between the signalling changes and the cognitive change.
2.2 (line 74) Experimental design: please explain the source and preparation of Mumefural (MF).
3.1. (line 206) please explain here briefly the UCCAO intervention and the rationale behind it as the rest of the article depends on this model. Please refer to Figure 1.
3.2 (line 237) the conclusion “Thus, these results suggest that MF improves UCCAO-induced cognitive dysfunction” is too general as the only function that improved was the recognition index (Fig. 3H).
3.3 (line 254) please introduce here CCH with its full name chronic cerebral hypoperfusion (CCH)
3.3. please comment in the main text on the impact of MF in the UCCAO model on the increase in hippocampus weight as it is the strongest effect
Figure 5A and 5B: it is difficult to judge whether the decrease in GLUT1, GLUT3 and AChe in the UCCAO model upon treatment with MF is significant. In the case of AChe the Western blot is not very convincing (Fig. 5B). I think it is difficult to judge this from a single sample experiment as it is unclear whether all protein samples are in the liner range and not within saturation. It would be better to show a dilution range starting with the same amount of total protein. If the expression of a protein within the total protein changes, this will be much easier to see in a dilution series compared to a single sample.
Figure 6B: please comment on the different phosphorylation levels of TrkB.
3.7 I feel that the protein expression analysis needs to be connected to the rest of the paper. For example, based on the data shown in Fig 6A, the protein expression of BDNF should differ significantly between UCCAO plus/minus MF. This should also be detectable in the proteomics data set. Please comment on this.
Given the strong change in BDNF, it may be worth exploring his protein more thoroughly. Please comment on the signalling role of BDNF in the Discussion in greater detail.
4. Discussion: please refer to the display items as well in the discussion
Only moderate editing required
Round 2
Reviewer 3 Report
Dear authors,
thank you very much for the thorough response to my coments. Prior to publication, I would like to ask you to add your good explanation why the proteins you identified by Western blot did not show up in the proteomics experiment
" Unfortunately, however, ERK, CREB, and BDNF proteins were not identified in the detection. Proteomics technologies are useful analysis systems to investigate biological function, but in the cases of the brain, neuroproteomics has probably not yet resulted in the detection of the majority of important brain proteins due to low abundance and hydrophobic properties (Reference A). For that reason, it is sometimes difficult to see results from Western blots in proteomics. However, we first showed that there were functional differences (e.g. neurodegeneration, oxidative phosphorylation) induced by UCCAO and MF treatment in the proteomic analysis"
Feel three to re-phrase it. I think it is important for the reader to know about this as one would expect to find proteins like BDNF in the proteomics study. You can include this information between lines 449 and 451.
Minor editing